# Regenerating Saffron (*Crocus sativus* L.) from Corm Lateral Buds via Indirect Somatic Embryogenesis

**DOI:** 10.3390/plants13010010

**Published:** 2023-12-19

**Authors:** Yangyang Ma, Yiyu Pan, Bizeng Mao

**Affiliations:** Institute of Biotechnology, Zhejiang University, Hangzhou 310058, China; mayangyang91@sina.com (Y.M.); 21816166@zju.edu.cn (Y.P.)

**Keywords:** *Crocus sativus*, differentiation, morpho-histological, somatic embryogenesis

## Abstract

Saffron (*Crocus sativus* L.), being one of the distinguished commercial spice crops in the world, is in demand for its culinary, colorant, and pharmaceutical benefits. In this study, a novel indirect somatic embryogenesis (SE) system was, thus, established for the study of this plant. To this end, firstly, the lateral buds were cultured. Then, the cultures were transformed using Murashige and Skoog (MS) medium supplemented with either 6-benzyladenine (BA: 5 and 10 mg/L), naphthalene acetic acid (NAA: 0, 1, and 2 mg/L), or trans-zeatin (tZ: 0, 0.5, and 1.0 mg/L), before being classified into four structures: white globular (WG), yellow compact nodular (YCN), yellow-brown fragile (YBF), and dark-brown porous (DBP). As soon as BA (10 mg/L) and NAA (2 mg/L) were added, elevated percentages of white globular calli (56.8%) and white globular calli (31.5%) structures were induced. Additionally, 6-benzyladenine (5 mg/L) and naphthalene acetic acid (1 mg/L) allowed the formation of yellow-brown fragile structures, and the combination of 6-benzyladenine (10 mg/L) with trans-zeatin (1 mg/L) formed the DBP structures. After three months, the white globular calli were incubated using the MS basal medium, before being augmented with thidiazuron (TDZ: 1 mg/L) and picloram (PIC: 2 mg/L), from which 60% of the cases matured into shoots and, ultimately, cormlets. Morphoanatomical analyses also showed that the white globular calli cells were closely arranged, as they had a dense cytoplasm, a significant vascular differentiation, and embryoids. Furthermore, the yellow compact nodular structures were characterized by a strong differentiation capacity and contained many meristematic cells with high caryomitosis centers. We observed that the yellow-brown fragile calli had looser cell arrangements, with a vascular structure located on the protoderm edge, while there was no obvious cellular arrangement in the dark-brown porous structures. The induction of the adventitious buds in vivo on the MS medium that was supplemented with thidiazuron and picloram accordingly demonstrated the highest rates (60%) of white globular calli.

## 1. Introduction

For centuries, saffron (*Crocus sativus* L.) has been cultivated to enable the use of its crimson stigmas and styles as seasonings and coloring agents in food, as well as for therapeutic purposes [1]. Accordingly, dehydrated saffron stigmas and styles are widely utilized in various industries, making them the world’s most expensive spice by weight [2]. Saffron flower components also contain more than 150 volatile and aroma-yielding compounds, as well as many non-volatile active compounds, such as carotenoids (e.g., α- and β-carotenes), zeaxanthin, and lycopene [3]. Domesticated saffron flowers, which produce the spice’s stigmas and styles, also occur in a sterile triploid form, meaning that they do not produce viable seeds or reproduce without human assistance.

Due to its asymmetrical meiosis, saffron pollen is non-viable. Conventional propagation systems, including digging up, cutting, and replanting corms, are, thus, insufficient and time-consuming; however, performing proliferation using tissue cultures can potentially make this process more efficient. Plant tissue cultures achieve plant multiplication via a rare process known as somatic embryogenesis (SE). The key feature of SE is the use of cellular totipotency, through which cell proliferation occurs based on dedifferentiation induction, followed by the subsequent induction of differentiation via the use of plant growth regulators to produce new plants [4]. Therefore, two SE pathways are present in plants: direct and indirect embryogenesis. Direct SE directly forms embryos from isolated cells without first forming callus tissue. In comparison, indirect organogenesis involves two steps. Firstly, the explants form calli, before enabling differentiation in vitro. Callogenesis then stimulates the explants and transforms them into a cluster of amorphous calli, which, in turn, transforms into defined organs under suitable conditions [5]. SE also occurs in calli, producing plantlets. In saffron, SE can be completed via the use of shoot meristems using Linsmaier and Skoog (LS) medium, supplemented with 2 μM of 6-benzyladenine (BA) and 2 μM of 1-naphthylacetic acid (NAA) [6]. Somatic embryo development has also been observed. This type of SE was reported by Blazquez [7], who employed 0.5 mg/L of BA and 0.1 mg/L of 2,4-dichlorophenoxyacetic acid (2,4-D) using MS medium to induce somatic embryos. Karamian [8] proposed inducing globular somatic embryos from shoot tips by supplementing the LS medium with 1 mg/L of 2,4-D and 4 mg/L of kinetin or 4 mg/L of NAA and 4 mg/L of BA, before transferring the germinated embryos into a half-strength MS medium containing 1 mg/L of BA and 1 mg/L of NAA to regenerate the plantlets. 

Moreover, Mahmud [9] analyzed the differences between the embryogenic and non-embryogenic calli of sugarcane (*Saccharum* sp.), finding that the metabolic profiles significantly differed in both types of calli. A similar report was provided regarding mangosteen by Maadon [10]. However, to the best of our knowledge, there is no research on saffron calli. In this study, the number of plant growth regulators (PGRs) was, thus, optimized, a high-efficiency regeneration system was established for saffron, and four types of structures were observed and named. This study represents the first analysis of the indirect organogenesis of saffron, wherein we differentiated the embryogenic and non-embryogenic calli from the lateral bud cultures via morphology and histology.

## 2. Results

### 2.1. Primary Calli Were Induced from the Lateral Bud Explants

The initial induction of the primary calli was conducted at the epidermis of the lateral buds (Figure 1) 25–30 days after inoculation. The number of calli induced depended on the number of PGRs in the initial MS medium. The highest callus production frequency of 86% (Table 1) was also induced from the lateral bud explants using NAA (0.2 mg/L) and BA (1.5 mg/L). The low concentrations of BA and NAA did not promote the culturing of the calli. When 0.2 mg/L was added to the MS medium, the calli’s formation frequency became higher than that for 0.1 mg/L NAA. The frequency of calli production by explants also increased as the BA concentration was increased, but the excessively high concentrations harmed explant growth.

### 2.2. Five Types of Calli Were Regenerated from Different Embryo Regeneration Media

After the eight-week cultivation in different embryo induction media, the calli developed into various structures: white globular (WG) (Figure 2A), yellow compact nodular (YCN) (Figure 2B), yellow-brown fragile (YBF) (Figure 2C), and dark-brown porous (DBP) (Figure 2D) structures. Other structures, such as the shoots or roots, were also formed in the embryo regeneration medium (Figure 3A). Depending on the concentration of the PGRs, the frequency of the structures varied. The highest percentages of WG (56%) and YCN (31%) calli were also obtained using the MS medium containing 10 mg/L of BA and 2 mg/L of NAA. In this sense, tZ did not promote the formation of the WG and YCN calli, and significant differences were noted between the tZ and NAA treatments. Treatment with BA (5 mg/L) and NAA (1 mg/L) was also the best method for developing the YBF calli. The YBF frequency was significantly diminished as NAA or tZ was added. The BA and tZ combination also produced a high percentage of DBP calli, with the greatest number recorded for the combination of 10 mg/L of BA and 1 mg/L of tZ. Furthermore, structures such as the shoots or roots were induced via these eight treatments. No differences were detected between the roots and shoots via various treatments (Table 2).

#### 2.2.1. Embryogenic Calli Were arranged Closely under Different Histological Observations

DAPI is a type of staining widely used for nucleus detection [11]. The DAPI staining showed the structures’ unique arrangements. The WG structures were the most closely arranged structures and showed strong cellular activity, while the DBP was more loosely arranged than other structures (Figure 2E–H). 

Moreover, borax-toluidine blue staining indicated different shades for the structures and cell arrangements. The WG cells were dense, stained dark blue, and had homogeneous cytoplasm. The DBP cells were much looser than other cells, staining red-purple in color (Figure 2I–L).

We also used the semi-thin sections to observe the differentiation in the morphology of various calli. The WG structures showed closely packed cells and scattered vascular bundles. The embryoids were also observed in the WG cells (Figure 2M), and, thus, the given structure was defined as embryogenic in nature. The YCN cells were dense, and meristematic cells and vascular tissues were observed (Figure 2N). This structure was also determined to be embryogenic in nature. The YBF callus cells were loosely arranged, the cells were large, a small number of epidermal cells were arranged neatly, and vascular bundle differentiation occurred (Figure 2O). Based on these factors, the YBF structure was also assumed to be embryogenic in nature. The DBP structures indicated large cell gaps and irregular cell shapes (Figure 2P). Moreover, there were no obvious nuclear structures or meristem cells. In the subsequent subculture, these loose cells gradually died. The unknown structure was the roots (Figure 2Q,R). 

The meristematic cells and the tracheary element differentiation present in the WG, YCN, and YBF calli demonstrated the presence of embryogenic cells. The DBP structures, which did not display any characteristics of embryonic cells, were ultimately hypothesized to be adult structures.

#### 2.2.2. Different Structures Show Different Surface and Internal Structures under TEM and SEM Observations

Based on the SEM outcomes, the WG calli had evident globular protrusions (Figure 3A) with smooth surfaces (Figure 3B), while the YCN calli surfaces were rough (Figure 3D). At a higher magnification, granular pellets were observed (Figure 3E). The DBP structures became dehydrated (Figure 3G) when pre-processed; nevertheless, the non-dehydrated parts showed a cluster of globular granules (Figure 3H). 

Likewise, the TEM analysis revealed WG cells with a dense protoplast, forming a dome containing rich starch grains, numerous mitochondria, and the Golgi apparatus (Figure 3C). These cells were thin-walled, with highly condensed nuclei, but the vacuoles were small or absent. The YCN calli also had many mitochondria and starch grains, as well as clear microtubule strands running through the middle of the cells and two complete chromosomes, which could give the most significant evidence of the high mitosis ability of these cells (Figure 3F). The YBF structures also had a large central vacuole, with metabolites (Figure 3I) and several ribosomes, the Golgi apparatus, and mitochondria (Figure 3J,K) present on the edge of the adjacent cells.

### 2.3. Shoots and Bulblet Were Produced under Higher Concentrations of PIC

After two months, the embryogenic callus was transferred to the shoot induction medium, and the bulb-like structures were observed. Histological observations revealed that the structures were shoots (Figure 4A). The shoot apical meristem (Figure 4B) was also observed, and the bulblet was formed and the scaly leaf was observed after the two-month culture (Figure 4C). Finally, multiple plantlets were formed from lateral buds (Figure 4D–G). During this process, different combinations of the plant hormones were added to the MS medium. The results are illustrated in Table 3. PIC produced the clustered buds when used with TDZ or BA, and there were no obvious differences between the treatments. However, the higher concentrations of PIC were more conducive to the induction of the clustered buds. When BA treatment alone was used, the induction rate of the clustered shoots was significantly lower than those of the other treatments. Therefore, 1 mg/L and 2 mg/L of PIC were the measures chosen to induce the clumping of the buds.

## 3. Discussion

Plant cells use cellular totipotency to produce new plantlets via SE or organogenesis [12]. In recent years, SE has been widely studied in plant species as a potential tool for rapid plant regeneration [13], especially for plants with long growth periods or low reproduction rates. Two somatic embryogenesis pathways exist in plants: direct and indirect embryogenesis. Various types of saffron explants are currently exploited via direct and indirect callogenesis, including juvenile parts, such as terminal/apical buds [14]; the shoot apical meristems with leaf primordia [4,6]; the sections derived from the central meristematic region [15]; and entire flower buds [16,17]. To the best of our knowledge, this study represents the first attempt to use lateral buds as explants. These bulbs produce a number of lateral buds (Figure 4A); therefore, they can be used to perform saffron multiplication.

PGRs are vital to callus induction and SE. BA are, thus, the best cytokinin supplements, when combined with NAA or 2,4-dichlorophenoxyacetic acid [18], for inducing calli. In this study, when a higher concentration of BA was combined with a higher concentration of NAA (Table 1), more calli were induced. Moreover, the calli induced from different PGR combinations were similar. SE could also allow the differentiation of a single somatic cell into an embryo or a plant. In the present study, different types of SE were induced via the MS medium to which different combinations of phytohormones had been added. The combination of auxin and cytokinin better promoted the initiation of SE [19]. In bulbous plants, BA was always used for in vitro embryogenesis of plants such as *Eucharis*, *Hippeastrum hybridum*, and *Crinum asiaticum* [20]. Moreover, other cytokinins, such as 6-furfurylaminopurine (kinetin), 2-imminopurine (2-ip), and tZ, were also reported to induce embryos in a single or combined manner [21,22]. However, the mechanism driving cytokinin-induced embryogenesis was unclear [4]. Therefore, in our study, we conducted embryo induction using the cytokinin combination of BA and tZ to provide evidence of embryo induction through the combination of cytokinins. However, the combination of BA and tZ was not suited to inducing embryogenic callus. The high induction and proliferation ratio for globular calli was 10 mg/L of BA combined with 1 or 2 mg/L of NAA without tZ (Table 2), suggesting that the combination of BA and NAA is superior to that of BA and tZ. Moreover, BA plays a dominant role in saffron embryogenic callus induction or proliferation, while tZ acts as a negative regulator. However, in other species, such as *Echinodorus orisis* L., the combination of BA (1 mg/L) and tZ (1 mg/L) induced the highest efficiency (100%) of somatic embryogenesis [23]. These results demonstrate that the mechanism driving SE differs between diverse species. 

Globular calli were similar to those observed by Devi [19] in in vitro saffron leaf cultures. The globular calli obtained from the cultures of lateral buds resembled somatic embryos, representing the most advanced embryogenic stage reached in this study. Histological, TEM, and SEM observations also supported the belief that such structures were somatic embryos. Moreover, Xu [24] reported that the embryogenic cells of *Solanum nigrum* were densely arranged based on staining with DAPI and borax-toluidine blue, and the same conclusion was reached based on analysis of the somatic embryos of saffron. The WG calli (Figure 2M) had meristematic cells (arrows), elongating the cells of tracheid differentiation, in addition to those of the vascular area, and the formation of tracheal elements with secondary walls. These findings are in accordance with those observed for globular embryos of soybean [25] and mangosteen [10]. The meristematic features described in the WG calli here showed high rates of cell division, dense cytoplasm, small vacuoles, Golgi apparatus, and numerous mitochondria based on the TEM observations (Figure 2C), thus corroborating the hypothesis of SE in the WG calli. 

The YCN calli were further induced using the same medium as the WG calli. These structures were similar to those observed by Devi [19] in saffron and other species, such as oil palm [26], *Brachypodium distachyon* [27], and mangosteen [10]. According to these reports, the given structures were thought to be embryogenic in nature. The SEM observations also provided evidence to support the presence of small WG calli on the surface of the nodular structure (Figure 3D,E). The findings revealed that the WG calli could later be induced from these structures. The TEM observations also showed that the YCN callus cells had strong mitosis, a large number of starch granules and microtubules, and nuclear materials, but the vacuole was small or absent. The starch granules and mitochondria were the sources of energy, supporting intense metabolic and mitotic activity, thus playing leading roles in morphogenesis. These results were consistent with the reports of other plants, such as *Quercus robur* [28] and *Curcuma mangga* [29].

The YBF calli (Figure 2C) were also formed using the MS medium with BA (5 mg/L) and NAA (1 mg/L). Such structures were loosely arranged and had little cell nuclei; at the same time, cellular activity was strong in some areas. The white square in Figure 2G shows a cell undergoing nuclear division. The SEM observations also revealed a few clusters of WG calli on the surface of this structure (Figure 3G,H). The volumes of vacuoles in the cells were large, and the content of the nucleus was small, but a few mitochondria, the Golgi apparatus, and ribosomes remained present based on the TEM images (Figure 3I,J,K). The histological observations also indicated that the vascular bundles could still be observed, though there was a large cell gap (Figure 2K). Therefore, these cells were differentiated and embryogenic in nature. In the saffron callus culture, the report on this structure was the first example considered. This structure appears to be non-embryogenic in *Dendrocalamus hamiltonii*, according to Zhang [30]. The reason for this discrepancy was attributed to the presence of different plant species and PGRs. Porous structures (Figure 2D) were also formed through treatment with 5 mg/L of BA and 0.5 mg/L of tZ. Few cytological features were observed. The DBP calli were shown to be non-embryogenic in nature, and they died at a later stage. The yellow-brown color was also caused by the oxidation and polymerization of the phenolics in the cytoplasm, as reported by Rai [31] in *Ipomea aquatica.*

In many plants’ bulbs, the conversion of somatic embryos into plantlets is difficult. A prior study reported that the lack of maturation and desiccation tolerance were the main factors driving low plant induction rates [32]. In this study, once the WG calli were transferred into the shoot induction medium, the shoots formed (Figure 4A), and the best shoot formation was the MS formation with 1 mg/L of TDZ and 2 mg/L of PIC. In this respect, Victor [33] reported that TDZ promoted purine metabolite accumulation by increasing the number of purines available for the development of cells. Once the corms formed, protein synthesis and cell division rapidly occurred, and TDZ was required for augmentation. However, at higher concentrations of TDZ, the rates of synthesis and division declined, likely due to the browning and death of the somatic embryos [23]. PIC is always used for SE [34]; however, it is rarely used in plantlet formation. However, our results suggested that PIC promotes the formation of shoots. Moreover, a certain ratio of TDZ and PIC significantly aids shoot formation in saffron. Therefore, in this study, we established the indirect somatic embryogenesis of saffron using lateral buds. Our findings provide a new line of research into the mass production and industrialized seedling culturing of saffron.

## 4. Materials and Methods

### 4.1. Plant Materials and Explant Preparation

Mature saffron (*Crocus sativus* L.) lateral buds (Figure 5A) were collected from some mature bulbs (Figure 5B), soaked in washing-up liquid for 10 min, and rinsed with water. The buds were also surface-sterilized for 1 min using 75% (*v*/*v*) ethanol, rinsed twice with sterilized water, treated in 5% (*v*/*v*) sodium hypochlorite (NaClO) for 20 min, and rinsed five times with sterilized water. The buds were then cut into several segments of 1.0 ± 0.5 cm in length.

The sterilized lateral bud segments were subsequently grown using an MS medium (1962) with BA (Sigma-Aldrich, Saint Louis, MO, USA) (0.5, 1.0, and 1.5 mg/L) and NAA (Sigma-Aldrich, Saint Louis, MO, USA) (0.1 and 0.2 mg/L), as well as sucrose (Aladdin, Los Angeles, CA, USA) (30 g/L) and phytagel (Coolaber, Beijing, China) (2.5 g/L). The medium pH was further lowered to 5.8–6.0 before being placed in the autoclave at 121 °C for 20 min. Three replicates were used for each treatment, as well as five orchid bottles for each replicate and three to five explants for each orchid bottle (650 mL bottles that have a smaller bottleneck than normal tissue culture bottles). The cultures were also incubated at 25 ± 1 °C in the dark for four weeks. The primary callus numbers were counted after four weeks, and the induction rate for each treatment was calculated (induction rate = primary callus number/explants number × 100%).

### 4.2. Somatic Embryogenesis Induction

All the cultures induced from the lateral buds were utilized for organogenesis or SE. The cultures were inoculated using the MS medium containing BA (5, 10 mg/L), NAA (0, 1, 2 mg/L), and trans-zeatin (tZ) (Sigma-Aldrich, Saint Louis, MO, USA) (0, 0.5, and 1 mg/L), as well as phytagel (2.5 g/L) and sucrose (30 g/L). Moreover, three replicates were used for each treatment, as well as five orchid bottles for each replicate and three to five cultures for each orchid bottle. The cultures were then observed every week, being transferred onto the fresh MS medium every four weeks. The different structures of the somatic embryos were also photographed using a digital camera (D750, Nikon, Japan). The cultures were then kept in the medium for eight weeks, and the numbers of each structure derived from different media were collected and the induction rates were calculated (induction rate = different structure number/primary callus number × 100%).

### 4.3. Shoot Regeneration and Bulblet Production

The resulting globular calli were relocated to the MS medium with picloram (PIC) (Sigma, USA) (0, 1, and 2 mg/L), BA (0 and 3 mg/L), or thidiazuron (Sigma-Aldrich, Saint Louis, MO, USA) (TDZ) (0 and 1.0 mg/L), as well as sucrose (30 g/L), for shoot regeneration. Three replicates were used for each treatment, as well as five 650 mL orchid bottles (bottles which have a smaller bottleneck than normal tissue culture bottles) for each replicate and three to five globular structures for each orchid bottle. The shoots were used for bulblet production, before being cultured on the MS medium (pH 5.8–6.0) with NAA (1 mg/L), agar (7.5 g/L), and sucrose (30 g/L). Notably, the shoots and cormlets were both cultivated for a photoperiod of 16 h at a constant temperature of 25 °C. After 30 days, the number of shoots was counted, and the regeneration rate was calculated (regeneration rate = shoot number/globular callus number × 100%).

### 4.4. Histological Analyses

To confirm the morphological features of the differentiating embryogenic structures, light and scanning electron microscopy (SEM) and transmission electron microscopy (TEM) were carried out. Different types of embryogenic structures were also selected and identified via SEM and TEM.

#### 4.4.1. DAPI and Borax-Toluidine Blue Staining

To observe the nuclei of the cells of both embryonic and callus tissues, the samples were randomly collected throughout the eight-week culturing stage. They were then stained with 4, 6-diamidino-2-phenylindole (DAPI) (Sigma, USA) by following a published method [24]. Next, the samples were kept in the liquid phosphate buffer solution (PBS (0.1 mol/L, pH 7.0); DAPI (5 μg/mL): PBS (*v*/*v*) = 1:1000) in darkness for 20 min. The freehand section was prepared using a scalpel, and a single layer of cells was then arranged on slides and photographed in the dark-field illumination of a confocal light microscope (BX 61, Olympus, Japan). Borax-toluidine blue staining, as described by Xu [35], was utilized to observe the cell outlines. The fresh materials were also soaked for 5 min in the borax-toluidine blue (Macklin, Shanghai, China), rinsed with sterilized water five times to wash away the floating dye, and dried with a filter paper to remove moisture. The microscopic images were then photographed under light-field illumination using a light microscope (BX 61, Olympus, Japan).

#### 4.4.2. Observation of Semi-Thin Sections under a Light Microscope

The samples (obtained during eight weeks) were fixed with 2.5% glutaraldehyde in the PBS (0.1 M, pH 7.0) for 12 h, washed with PBS (0.1 M, pH 7.0), and post-fixed with 1% osmium tetroxide (OsO_4_) in phosphate buffer (0.1 M, pH 7.0) for 1–2 h. Following fixation, a graded series of ethanol (30, 50, 70, 80, 90, 95, and 100%) was utilized to dehydrate the samples for 15–20 min at each step. The samples were then transferred to absolute acetone for 20 min and infiltrated in a 1:1 mixture of acetone, and a final Spurr resin mixture was maintained for 1 h at room temperature. The samples were then transferred to a mixture of acetone and the final resin mixture at a ratio of 25:75 for 3 h, before being transferred to the Spurr resin mixture overnight. Such samples were placed in centrifuge tubes containing Spurr resin and heated at 70 °C for 9 h. They were also sliced to create sections in a thin-slice machine (LKB 11800, PYRAMITOME, Sweden), wherein methylene blue (Aladdin, Los Angeles, CA, USA) was used for staining. After staining the tissue with methylene blue for 5 min, the float was rinsed off with double-steamed water. The images were finally captured using an optical microscope (ECLIPSE Ni-U, Nikon, Japan).

#### 4.4.3. Scanning Electron Microscopy (SEM) Observation

To further compare the cytological differences between the calli and different structures, transmission and scanning electron microscopy were performed. The calli of different structures were selected and divided into sections of 3 mm × 3 mm. The samples were fixed with 2.5% glutaraldehyde for longer than 4 h in PBS (0.1 M, pH 7.0), washed with the same buffer, post-fixed with 1% OsO_4_ in phosphate buffer (0.1 M, pH 7.0) for 1–2 h, dehydrated for 15–20 min for each step in a graded series of ethanol (30, 50, 70, 80, 90, 95, and 100%), and transferred to a 1:1 mixture of alcohol and iso-amyl acetate for 0.5 h, before being transferred to pure iso-amyl acetate for 1 h. The samples were also dehydrated in an HCP-2 critical point dryer (HCP-2, Hitachi, Japan) with liquid carbon dioxide (CO_2_). Further, gold–palladium was utilized to coat the dehydrated samples in an E-1010 Ion Sputter (E-1010, Hitachi, Japan) for 4–5 min, which we observed using a Model SU-8010 SEM (SU-8010, Hitachi, Japan).

#### 4.4.4. Transmission Electron Microscopy (TEM) Observation

The preparation for TEM followed a similar procedure to that used in the processing of semi-thin sections, including the steps of double fixation, graded dehydration series, infiltration, and embedding. An Ultramicrotome Leica EM UC7 (EM UC7, Leica, Germany) was then employed to divide the embedded samples into sections, and the resulting sections were stained with alkaline lead citrate and uranyl acetate for approximately 10 min, before being observed with an H-7650 TEM (H-7650, Hitachi, Japan).

### 4.5. Statistical Analysis

All data were statistically analyzed using the SPSS Statistics software package (ver. 19.0; IBM, Armonk, NY, USA). Data were also analyzed via Duncan’s new multiple range test (DMRT), and the mean values were separated (*p* < 0.05).

## 5. Conclusions

In conclusion, indirect SE was achieved by using the lateral bud explants of saffron, and the histological observations of SE were studied. The WG structure was the best material for the induction of adventitious buds. Eventually, the regenerated plantlets were successfully established via indirect SE. This study’s results, therefore, have implications for germplasm storage and the genetic enhancement of saffron.

## Figures and Tables

**Figure 1 plants-13-00010-f001:**
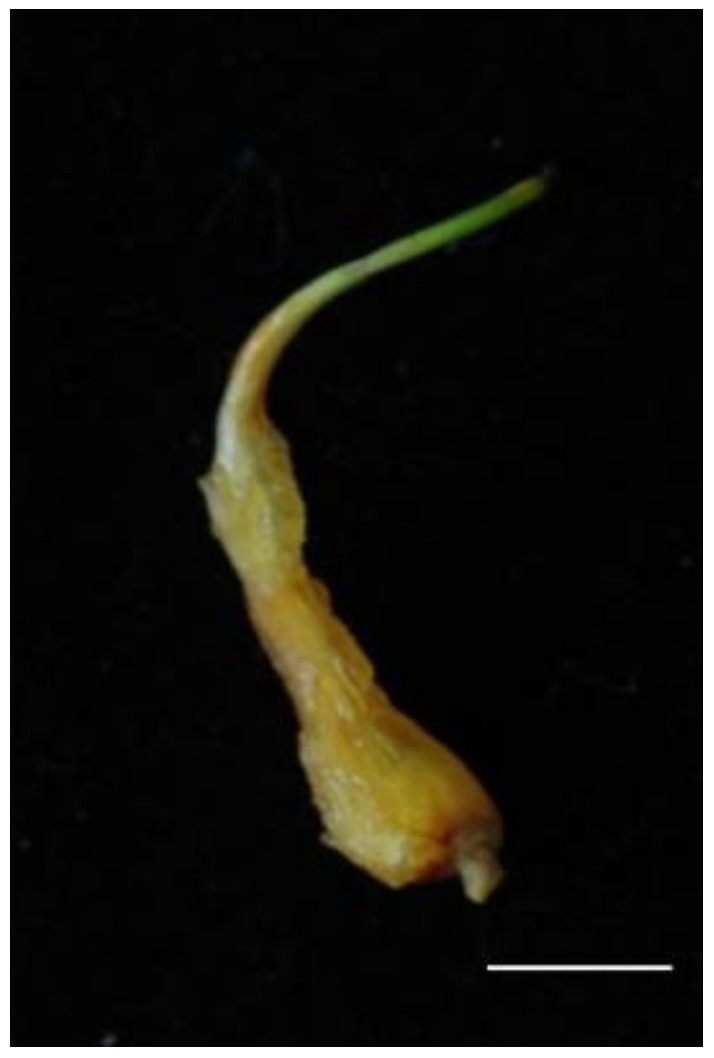
Primary callus induced from the epidermis of a lateral bud. Scale bar: 1 cm.

**Figure 2 plants-13-00010-f002:**
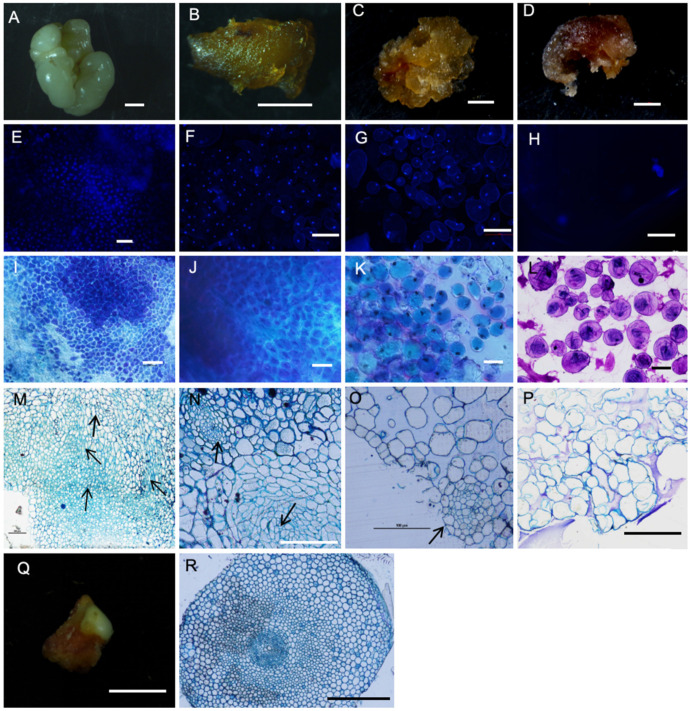
Four types of structures induced from saffron lateral buds. (**A**) White globular structure, scale bar: 1 cm; (**B**) yellow compact nodular structure, scale bar: 1 cm; (**C**) yellow-brown fragile structure, scale bar: 1 cm; (**D**) dark-brown porous structure, scale bar, 1 cm; (**E**) cell nuclei of globular structures stained with DAPI and observed via dark-field lighting, scale bar: 100 μm; (**F**) cell nuclei of nodular structures stained with DAPI and observed via dark-field lighting, scale bar: 500 μm; (**G**) cell nuclei of fragile structures stained with DAPI and observed via dark-field lighting, scale bar: 500 μm; (**H**) cell nuclei of porous structures stained with DAPI and observed via dark-field lighting, scale bar: 100 μm; (**I**) cell morphology and arrangement of globular structures stained with borax-toluidine blue, scale bar: 200 μm; (**J**) cell morphology and arrangement of nodular structure stained with borax-toluidine blue, scale bar: 200 μm; (**K**) cell morphology and arrangement of fragile structure stained with borax-toluidine blue, scale bar: 10 μm; (**L**) cell morphology and arrangement of porous structure stained with borax-toluidine blue, scale bar: 500 μm; (**M**) histological section of globular structure, in which the arrows represent meristematic cells, leading to elongated cells of tracheid differentiation, and the box areas represent the vascular area and the formation of tracheal elements with secondary walls and distinct protoderm and tracheary differentiation, the arrow shows meristematic cells. scale bar: 100 μm; (**N**) histological section of the nodular structure with meristematic cells and vascular bundle differentiation, the arrow shows vascular bundles. scale bar: 100 μm; (**O**) histological section of the fragile structure with neat epidermal cells and vascular tissue differentiation, the arrow shows vascular bundle. scale bar: 100 μm; (**P**) histological section of the porous structure, scale bar: 100 μm; (**Q**) root, scale bar: 1 cm; (**R**) histological section of the unidentifiable structure found to be roots, scale bar: 50 μm.

**Figure 3 plants-13-00010-f003:**
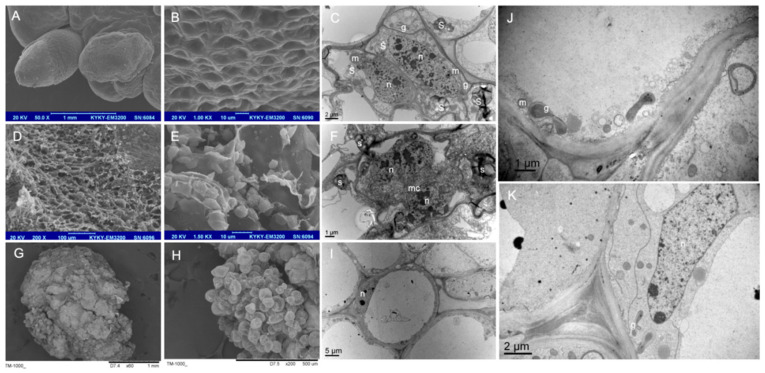
SEM and TEM observations of different structures. (**A**) Scanning electron micrograph of the WG structure; (**B**) magnification of (**A**) with a smooth surface; (**C**) transmission electron micrographs of the WG structure; (**D**) scanning electron micrographs of the YCN structure; (**E**) magnification of D with a rough surface; (**F**) transmission electron micrograph of the YCN; (**G**) scanning electron micrograph of DBP; (**H**) magnification of the box area in G with a cluster of globular granules; (**I**) transmission electron micrograph of DBP; (**J**,**K**) details of fragile structures determined through transmission electron micrographs; (s) starch grain; (n) nucleus; (g) Golgi apparatus; (m) mitochondria; (v) vacuole; (mc) microtubule.

**Figure 4 plants-13-00010-f004:**
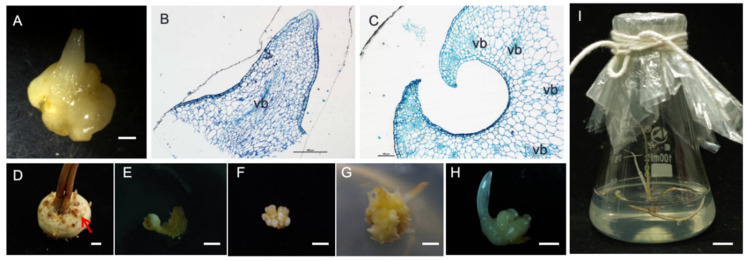
The regeneration process of *Crocus sativus* was performed via indirect somatic embryogenesis: (**A**) bulb-like structure; (**B**) the shoot apex of a bulblet; (**C**) the structure of a scale leaf; (**D**–**H**) the formation of multiple plantlets from an in vitro WG; (**I**) a regenerated plant; (vb) vascular bundle. Scale bar: (**A**) 1 cm; (**B**) 100 μm; (**C**) 500 μm; (**D**–**I**) 1 cm.

**Figure 5 plants-13-00010-f005:**
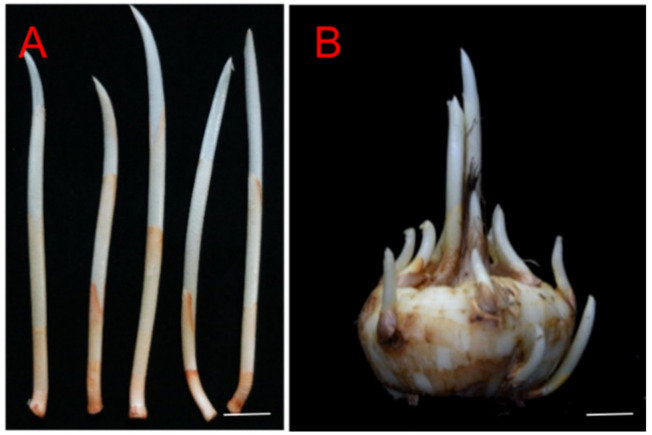
The mature saffron lateral buds and bulbs. (**A**) *Crocus sativus* lateral buds; (**B**) mature bulb. Scale bar: 1 cm.

**Table 1 plants-13-00010-t001:** Percentage of lateral bud response after culturing using the MS medium with the treatment combinations and concentrations of BA and NAA.

BA	NAA	Explants Showing Callus Induction Rate (%)
0.5	0.1	25.1 ± 0.0 b
0.5	0.2	56.3 ± 21.2 a
1	0.1	49.7 ± 22.4 ab
1	0.2	86.5 ± 2 a
1.5	0.1	50 ± 0.00 b
1.5	0.2	71.5 ± 10.4 a

Note: mean values with different letters are significantly different at *p* < 0.05 based on Duncan’s new multiple range test (DMRT).

**Table 2 plants-13-00010-t002:** Development of different structures for 8 weeks after culture using MS medium with treatment combinations and concentrations of BA, NAA, and tZ.

BA (mg/L)	NAA (mg/L)	tZ (mg/L)	WG (%)	YCN (%)	YBF (%)	DBP (%)	Roots (%)	Shoots (%)
5	1	0	25.0 b	30.6 a	41.4 a	0 b	1.8 c	1.2 c
5	2	0	42.3 ab	21.0 ab	19.0 b	13.7 ab	3.1 c	0.9 c
5	0	0.5	13.9 b	6.3 b	37.5 a	35.3 a	5.2 c	1.8 c
5	0	1	14.5 b	16.9 b	10.9 b	43.5 a	10.1 b	4.0 c
10	1	0	50.0 a	23.9 a	22.4 ab	0.1 b	2.6 c	1 c
10	2	0	56.8 a	31.5 a	10.7 b	0 b	0.8 c	0.2 c
10	0	0.5	10.9 b	4 b	38.8 a	48.0 a	0.7 c	1.13 c
10	0	1	25.0 b	0.2 b	0 b	47.4 a	0.6 b	27.1 a

Note: mean values with different letters within each column are significantly different at *p* < 0.05 based on Duncan’s new multiple range test (DMRT). NAA, naphthalene acetic acid; tZ, zeatin; WG, white globular; YCN, yellow compact nodular; YBF, yellow-brown fragile; DBP, dark-brown porous.

**Table 3 plants-13-00010-t003:** Percentage of buds induced using MS medium with the treatment combinations and concentrations of BA, TDZ, and PIC.

BA (mg/L)	TDZ (mg/L)	PIC (mg/L)	Bud Induction Rate (%)
0	1	1	35.2 ± 9 a
0	1	2	60.4 ± 16.5 a
3	0	0	13.5 ± 12.6 b
3	0	1	32.8 ± 8.7 a
3	0	2	41.3 ± 11.9 a

Note: mean values ± standard deviations with different letters are significantly different at *p* < 0.05 based on Duncan’s new multiple range test (DMRT).

## Data Availability

Original data is available upon request from the corresponding author.

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
