# Peer review of "Regenerating Saffron (Crocus sativus L.) from Corm Lateral Buds via Indirect Somatic Embryogenesis"

_plants, 2023, doi:10.3390/plants13010010_

Round 1

Reviewer 1 Report

Comments and Suggestions for Authors

The manuscript needs minor english revision. The introduction is sufficient and well written. Material section needs  additions of the details how certain data were calculated. Resulst needs improvements in the tables and Figures. Fig 3 needs to be carefully revised and changed. Discussion could be enriched with the brief summary of the way of inducing somatic embryogenesis presented by authors and the discussion of this pathway with the extremly rich literature on somatic embryogenesis in other species. 

  All detailed comments are in pdf. version.

Comments on the Quality of English Language

The manuscript needs minor english revision.

Author Response

Dear Editor

On behalf of all authors of this manuscript, I would like to thank you and the reviewers for taking time to review our manuscript and provide insightful comments.

We have addressed all the questions raised by the reviewers and editor and made appropriate changes/corrections accordingly, we also highlighted these revised texts in the new version of manuscript. All changes have been highlighted in light blue in the main text. Below are the specific responses to the questions.

If you have any questions, please do not hesitate to contact us. Thank you very much for serving as the editor for our manuscript. We look forward to hearing from you soon.

Best wishes,

Bizeng Mao

Responds to the reviewer’s comments:

Reviewer #1 (Comments for the Author):

  1. The manuscript needs minor english revision.

Response: We thank the reviewer#1’s comments. We have polished our manuscript.

  1. The introduction is sufficient and well written.

Response: Thank you for your comments. 

  1. Material section needs additions of the details how certain data were calculated.

Response: Thank you for your suggestions. We have add data calculated details in material and marked blue.

  1. Resulst needs improvements in the tables and Figures.

Response: Many thanks for your suggestions. We have re-written the tables and figures.

  1. Fig 3 needs to be carefully revised and changed.

Response: Thank you for your suggestions. We have rearranged the Fig 3.

  1. Discussion could be enriched with the brief summary of the way of inducing somatic embryogenesis presented by authors and the discussion of this pathway with the extremly rich literature on somatic embryogenesis in other species.

Response: We thank the reviewer#1’s comments. We have add the brief summary of the way of inducing somatic embryogenesis and somatic embryogenesis in other species.

Other changes that need to be made have been made by referring to pdf.

Reviewer 2 Report

Comments and Suggestions for Authors

In the submitted manuscript, the authors describe in vitro regeneration of saffron (Crocus sativus L.) from lateral buds via indirect somatic embryogenesis. Using different cultivation media and combination of plant growth regulators, they regenerated calli and subsequently somatic embryos, shoot and whole plants. The topic and methodology is promising with results that can be interesting for readers from biotechnology field. However, the manuscript would need substantial improvement both in terms of the text and the information included.

Suggestions:

1. In the abstract, it would be better not to use short forms.

2. As the manuscript is focused on somatic embryogenesis it would be useful to add the description into an introduction.

3. Figure 1: the scale bar can be written as 1 cm instead of 1000 micrometers.

4. In phytohormonal field, the “tZ” is usually used as a short form for a cytokinin trans-zeatin.

5. Can you define “orchid bottle”?

6. The chapter 2.4. van be excluded as a description of SEM and TEM is included in following chapters.

7. Composition of PBS buffer is missing.

8. Microscopy can be described in separate chapter.

9. Description of “methylene blue stainning” is missing.

10. Sample preparation for SEM and TEM should be described in more detail. The full name of SEM and TEM should be used in a chapter titles.

11. All results under different cultivation conditions (concentrations of PGRs) should be documented by representative images.

12. Figures 3-5: images should be named alphabetically (not A1, A2, A3 etc.).

13. The quality of the Figure 4 should be improved.

14. Chapter titles in “Results” should stress results, not the methodology.

15. The wording should be consistent in whole text: e.g., “WG calli” vs “WG structures”.

16. There are a lot of short forms in the text. The reduced number of short forms would make the text easier to read.

Comments on the Quality of English Language

English correction would improve the quality of the manuscript.

Author Response

Dear Editor

On behalf of all authors of this manuscript, I would like to thank you and the reviewers for taking time to review our manuscript and provide insightful comments.

We have addressed all the questions raised by the reviewers and editor and made appropriate changes/corrections accordingly, we also highlighted these revised texts in the new version of manuscript. All changes have been highlighted in light blue in the main text. Below are the specific responses to the questions.

If you have any questions, please do not hesitate to contact us. Thank you very much for serving as the editor for our manuscript. We look forward to hearing from you soon.

Best wishes,

Bizeng Mao

Reviewer #2 (Comments for the Author):

In the submitted manuscript, the authors describe in vitro regeneration of saffron (Crocus sativus L.) from lateral buds via indirect somatic embryogenesis. Using different cultivation media and combination of plant growth regulators, they regenerated calli and subsequently somatic embryos, shoot and whole plants. The topic and methodology is promising with results that can be interesting for readers from biotechnology field. However, the manuscript would need substantial improvement both in terms of the text and the information included.

We thank the reviewer#2’ s comments and suggestions. Below are the responses of previous concerns by the two reviewers with point-by-point.

  1. In the abstract, it would be better not to use short forms.

Response: We thank the reviewer#2 for this valuable comments and suggestions. We have checked our abstract and made ensure that all short forms had an explanation in previous text.

  1. As the manuscript is focused on somatic embryogenesis it would be useful to add the description into an introduction.
    Response: Many thanks for your suggestion. We have add some description about somatic embryogenesis in our introduction. You can find our revision in Line 44-54.
  2. Figure 1: the scale bar can be written as 1 cm instead of 1000 micrometers.

Response: Thank you for your comments and suggestions. We have instead the scale bar.

  1. In phytohormonal field, the“tZ”is usually used as a short form for a cytokinin trans-zeatin.

Response: We thank for your suggestion. However, we cannot find any “tZ” in our text.

  1. Can you define “orchid bottle”?

Response: Thanks. “ orchid bottle” is always used for tissue culture of orchids or medicines which were bigger than normal bottles used in tissue culture.  

  1. The chapter 2.4. van be excluded as a description of SEM and TEM is included in following chapters.

Response: Many thanks for your suggestion. We have re-written this chapter name.

  1. Composition of PBS buffer is missing.

Response: Thanks. We described the PBS we use in this study was pH 7.0, you can find in Line127.  

  1. Microscopy can be described in separate chapter.

Response: Thank you for your suggestion. We have re-written this part and you can find in Line 117-171.

  1. Description of “methylene blue stainning”is missing.

Response: Many thanks for your suggestion. We have added the description of methylene blue stainning, you can find in Line 148-150.

  1. Sample preparation for SEM and TEM should be described in more detail. The full name of SEM and TEM should be used in a chapter titles.

Response: Thanks. We have added the full name of SEM and TEM in the chapter title according to your suggestion. The sample preparation were described and you can find in Line 153-155.

  1. All results under different cultivation conditions (concentrations of PGRs) should be documented by representative images.

Response: Many Thanks for your suggestion. Because different combination of PGRs may produce similar structures of callus, we have classified the structures, and these structures were very small, we show separated structures other than summary image.    

  1. Figures 3-5: images should be named alphabetically (not A1, A2, A3 etc.).

Response: Thank you for your suggestion. We have rearranged the pictures per your comments.

  1. The quality of the Figure 4 should be improved.

Response: Thanks. We have replaced pictures in Figure 4.

  1. Chapter titles in “Results” should stress results, not the methodology.

Response: Thank you for your suggestion. We have rewritten the chapter titles according to your suggestion.

  1. The wording should be consistent in whole text: e.g., “WG calli” vs “WG structures”.

Response: We are sorry for our unclear description. We have revised our description to make it consistent and marked blue.

Round 2

Reviewer 1 Report

Comments and Suggestions for Authors

The manuscript is much better now, however the Authors did not refered to some of my questions - attached in the revised pdf.

Author Response

Dear Editor

On behalf of all authors of this manuscript, I would like to thank you and the reviewers for taking time to review our manuscript and provide insightful comments again.

We have addressed all the questions raised by the reviewers and editor and made appropriate changes/corrections accordingly, we also highlighted these revised texts in the new version of manuscript. All changes have been highlighted in light red in the main text to distinguish it from the first blue . Below are the specific responses to the questions.

If you have any questions, please do not hesitate to contact us. Thank you very much for serving as the editor for our manuscript. We look forward to hearing from you soon.

Best wishes,

Bizeng Mao

Responds to the reviewer’s comments:

Reviewer #1 (Comments for the Author):

I dont understand why these analyses were performed and what relevant informations it provided refering that Authors performed SEM and TEM.

We thank the reviewer#1’ s comments and suggestions. DAPI staining was conducted to visualize cell nuclei,according to the arrangement to cell nuclei and other histological staining,the embryogenesis cells and non-embryogenesis cells were distinguished. Other researchers have also used this method to study the arrangement of embryogenic callus. Relevant references are as follows:

Xu K, Huang X, Wu M, Wang Y, Chang Y, Liu K, Zhang J, Zhang Y, Zhang F, Yi L, Li T, Wang R, Tan G, Li C. A rapid, highly efficient and economical method of Agrobacterium-mediated in planta transient transformation in living onion epidermis. PLoS One. 2014 Jan 8;9(1):e83556. doi: 10.1371/journal.pone.0083556.

Xu KD, Chang YX, Zhang J, Wang PL, Wu JX, Li YY, Wang XW, Wang W, Liu K, Zhang Y, Yu DS, Liao LB, Li Y, Ma SY, Tan GX, Li CW. A lower pH value benefits regeneration of Trichosanthes kirilowii by somatic embryogenesis, involving rhizoid tubers (RTBs), a novel structure. Sci Rep. 2015 Mar 6;5:8823. doi: 10.1038/srep08823.  

The TEM and SEM were performed to find the ultrastructural differences because other histological staining cannot provide differences in organelle development and callus surface structure.

Other revision was made in our manuscript according to your sugge

Reviewer 2 Report

Comments and Suggestions for Authors

I appreciate authors’ effort to improve the manuscript. However, there are still some points that should be addressed.

Major comments:

1. All results described in the manuscript should be documented by representative images, e.g., lines 192-204. Roots and shoots should be evaluated separately as they represent organs regenerated by different developmental programs (Table2).

2. As the in vitro regeneration is regulated by a combination of phytohormones auxin and cytokinin, the usage of two cytokinins zeatin and BA together in one cultivation medium should be discussed.

3. The phytohormone zeatin used in tissue cultures is usually trans-zeatin (the cis-zeatin exists as well). It should be clarified whether authors used trans-zeatin or cis-zeatin. If trans-zeatin was used, the word “zeatin” should be written as “trans-zeatin” and the short form should be “tZ” in the whole text. The source of chemicals (company) would be useful to add into “Methods”.

Minor comments:

1. In the abstract, it would be better not to use short forms, if possible.

2. Lines 53-57:  “In saffron, SE can be completed by the use of shoot meristems on the Linsmaier and Skoog (LS) medium, supplemented with 2 μM 6-benzyladenine (BA) and 2 μM 1-naphthylacetic acid (NAA)[6]. Somatic embryo development has been observed, too. This type of SE was similarly reported by Blazquez[7], wherein 6-benzylaminopurine (BAP: 0.5 mg/L) and 2, 4-dichlorophenoxyacetic acid (0.1 mg/L) in the MS medium were employed to induce somatic embryos.”

- BA and BAP represent the same chemical. The same wording should be used in a whole text.

3. The full length of a short form should be written ones in the text when it appears for the first time.

4. There are misspelling in the text, they should be corrected.

5. The term “bar” should be replaced by “scale bar”.

6. The precise description what the “orchid bottle” means should be added into “Methods”.

7. Composition of PBS buffer should be added into “Methods”, not only pH.

Comments on the Quality of English Language

Minor edits are required.

Author Response

Dear Editor

On behalf of all authors of this manuscript, I would like to thank you and the reviewers for taking time to review our manuscript and provide insightful comments again.

We have addressed all the questions raised by the reviewers and editor and made appropriate changes/corrections accordingly, we also highlighted these revised texts in the new version of manuscript. All changes have been highlighted in light red in the main text to distinguish it from the first blue . Below are the specific responses to the questions.

If you have any questions, please do not hesitate to contact us. Thank you very much for serving as the editor for our manuscript. We look forward to hearing from you soon.

Best wishes,

Bizeng Mao

Responds to the reviewer’s comments:

Reviewer #2 (Comments for the Author):

Major comments:

  1. All results described in the manuscript should be documented by representative images, e.g., lines 192-204. Roots and shoots should be evaluated separately as they represent organs regenerated by different developmental programs (Table2).

Response:We thank the reviewer#2 for this valuable comments and suggestions. The focus of our manuscript was somatic embrgenesis and regeneration of saffron. Five bottles were used for each treatment, with five explants in each bottle, and a total of three treatments were conducted as the basis for statistical data. However, the growth type of callus in a single bottle was not representative, so it was not included in the manuscript. In addition, we found that there are no representative images of different processing types in other research of the same type, the references were below. We re-evaluated the roots and shoots rate separately according to your suggestion.

References: Avila-Victor, C.M.; Arjona-Suárez, E.d.J.; Iracheta-Donjuan, L.; Valdez-Carrasco, J.M.; Gómez-Merino, F.C.; Robledo-Paz, A. Callus Type, Growth Regulators, and Phytagel on Indirect Somatic Embryogenesis of Coffee (Coffea arabica L. var. Colombia). Plants 2023, 12, 3570. https://doi.org/10.3390/plants12203570

Ku, S.S.; Woo, H.-A.; Shin, M.J.; Jie, E.Y.; Kim, H.; Kim, H.-S.; Cho, H.S.; Jeong, W.-J.; Lee, M.-S.; Min, S.R.; et al. Efficient Plant Regeneration System from Leaf Explant Cultures of Daphne genkwa via Somatic Embryogenesis. Plants 2023, 12, 2175. https://doi.org/10.3390/plants12112175

  1. As the in vitro regeneration is regulated by a combination of phytohormones auxin and cytokinin, the usage of two cytokinins zeatin and BA together in one cultivation medium should be discussed.

Response: Thank you for your suggestion. We have added discussion about two cytokinins zeatin and BA and you can find in Line 354-362.

  1. The phytohormone zeatin used in tissue cultures is usually trans-zeatin (the cis-zeatin exists as well). It should be clarified whether authors used trans-zeatin or cis-zeatin. If trans-zeatin was used, the word “zeatin” should be written as “trans-zeatin” and the short form should be “tZ” in the whole text. The source of chemicals (company) would be useful to add into “Methods”.

Response: We are sorry for our unclear description. We have revised our description as “trans-zeatin” and added the source of chemicals in the methods.

Minor comments:

  1. In the abstract, it would be better not to use short forms, if possible.

Response: Thank you for your suggestion. We have provided full forms in the abstract.

  1. Lines 53-57:“In saffron, SE can be completed by the use of shoot meristems on the Linsmaier and Skoog (LS) medium, supplemented with 2 μM 6-benzyladenine (BA) and 2 μM 1-naphthylacetic acid (NAA)[6]. Somatic embryo development has been observed, too. This type of SE was similarly reported by Blazquez[7], wherein 6-benzylaminopurine (BAP: 0.5 mg/L) and 2, 4-dichlorophenoxyacetic acid (0.1 mg/L) in the MS medium were employed to induce somatic embryos.”

- BA and BAP represent the same chemical. The same wording should be used in a whole text.

Response: We are sorry for our unclear description. We have revised our description to make it consistent and marked red.

  1. The full length of a short form should be written ones in the text when it appears for the first time.

Response: We are sorry for our unclear description. We have revised our description to make it consistent and marked red.

  1. There are misspelling in the text, they should be corrected.

Response: We're sorry we didn't check it carefully. We have checked it.

  1. The term “bar” should be replaced by “scale bar”.

Response: Thanks. We have replaced it in our manuscript.

  1. The precise description what the “orchid bottle” means should be added into “Methods”.

Response: Thank you for your comments. We have add the description of orchid bottle and you can find in Line 114-115.

  1. Composition of PBS buffer should be added into “Methods”, not only pH.

Response: Many thanks for your comments. We have add the composition of PBS buffer in method and you can find in Line 132-133.

Round 3

Reviewer 2 Report

Comments and Suggestions for Authors

I appreciate authors’ effort to improve the manuscript. However, there are still some points that should be addressed before publishing.

Comments:

1. Results described in the manuscript should be documented by representative images.

2. Table 1: in case of chemicals, units should be added. It is not clear what does it mean that e.g., 0.25 of explant displayed the callus formation. Can authors clarify this in the table legend? How many samples were used? How was the number calculated?

3. BA and BAP represent the same chemical. The same wording should be used in a whole text, including long forms.

- still valid comment

4. The full length of a short form should be written ones in the text when it appears for the first time.

- still valid comment

5. There are misspelling in the text, they should be corrected.

- still valid comment

6. Composition of PBS buffer should be written with final concentrations of chemicals, not mass (g).

Comments on the Quality of English Language

See above.

Author Response

Dear Editor

On behalf of all authors of this manuscript, I would like to thank you and the reviewers for taking time to review our manuscript and provide insightful comments again.

We have addressed all the questions raised by the reviewers and editor and made appropriate changes/corrections accordingly, we also highlighted these revised texts in the new version of manuscript. All changes have been highlighted in light yellow in the main text to distinguish it from the first blue and red . Below are the specific responses to the questions.

If you have any questions, please do not hesitate to contact us. Thank you very much for serving as the editor for our manuscript. We look forward to hearing from you soon.

Best wishes,

Bizeng Mao

Responds to the reviewer’s comments:

Reviewer #2 (Comments for the Author):

Major comments:

  1. Results described in the manuscript should be documented by representative images.

Response:We thank the reviewer#2 for this valuable comments and suggestions. We have re-written the results of Figure 3 and Figure 5, and you can find in Line 318-324.

  1. Table 1: in case of chemicals, units should be added. It is not clear what does it mean that e.g., 0.25 of explant displayed the callus formation. Can authors clarify this in the table legend? How many samples were used? How was the number calculated?

Response: Thank you for your suggestion. We have added units in Table 1 and data calculation methods. You can find in Line 94-96.

  1. BA and BAP represent the same chemical. The same wording should be used in a whole text, including long forms.

Response: We are sorry for our unclear description. We have revised our description as BA.

  1. 4. The full length of a short form should be written ones in the text when it appears for the first time.

Response: Thank you for your suggestion. We have provided full forms when it appears for first time.

  1. 5. There are misspelling in the text, they should be corrected.

Response: We are sorry for our mistake. We have corrected misspelling and marked yellow .

  1. 6. Composition of PBS buffer should be written with final concentrations of chemicals, not mass (g).

Response: Thank you for your suggestion. We have revised our description and you can find in Line 134.

Round 4

Reviewer 2 Report

Comments and Suggestions for Authors

I appreciate authors’ effort to improve the manuscript and the text. The authors replayed questions and fulfilled the edits. The manuscript can be published in "Plants" after minor English revisions.

Comments on the Quality of English Language

Se above.

Author Response

I appreciate authors’ effort to improve the manuscript and the text. The authors replayed questions and fulfilled the edits. The manuscript can be published in "Plants" after minor English revisions.

Response:Thank you. We have revised the English language of the manuscript.
